# Executive functioning is associated to everyday interference of pain in patients with chronic pain

Nils Berginström[1,2]*, Sofia Wåhlin[3], Linn Österlund[3], Anna Holmqvist[4], Monika Löfgren[4], Britt-Marie Stålnacke[1], Marika C. Möller[4,5]

1 Department of Community Medicine and Rehabilitation, Rehabilitation Medicine, Umeå University, Umeå, Sweden, 2 Department of Psychology, Umeå University, Umeå, Sweden, 3 Sundsvall-Härnösand County Hospital, Sundsvall, Sweden, 4 Department of Clinical Sciences, Division of Rehabilitation Medicine, Karolinska Institutet, Danderyd Hospital, Stockholm, Sweden, 5 Department of Rehabilitation Medicine Stockholm, Danderyd University Hospital, Stockholm, Sweden

* nils.berginstrom@umu.se

## Abstract

Dysfunction in executive functions is common among patients with chronic pain. However, the relationships between executive functioning and pain management have not been extensively studied. In this study, 189 outpatients (160 women, 29 men; mean age 33.15) with chronic pain underwent an extensive neuropsychological assessment, including several tests of executive functions. In addition, all participants completed self-assessment questionnaires regarding pain and interference of pain in everyday life. After adjusting for effects of age, education, and depression, several aspects of executive functioning were significantly associated with self-assessed everyday interference of pain ($rs = 0.13$–$0.22$, all $ps < 0.05$). This indicates that lower performance on tests of executive functioning was significantly associated with a higher degree of pain interference and a lower degree of life control. Pain characteristics such as pain intensity, pain duration, and pain spreading were not associated with executive functioning. These results suggest that preserved executive functions are related to better coping with pain, but not directly to the pain itself, in patients with chronic pain. Depression was also associated with self-management of pain, indicating that patients with lower executive functioning in combination with depression may need special attention during rehabilitation.

## Introduction

Chronic pain, defined as pain persisting or recurring for longer than 3 months, is associated with significant emotional distress and functional disability [1]. The new definition of pain from the International Association for the Study of Pain [2] states pain to be an unpleasant sensory or emotional experience that can be, but not necessarily is, associated with actual tissue damage. The new definition and notes were revised to better explain the complexity of pain in order to better assess and manage people with pain. While acute pain plays a protective role by

**Data Availability Statement:** Anonimized data are available from the Zenodo repository (Zenodo.org) (DOI: 10.5281/zenodo.13809000).

**Funding:** This study was funded by Karolinska Institutet, Department of Clinical Sciences, the Promobilia Foundation (no. A22056), by the research and development fund granted by Västerbotten County Council, and through a regional agreement between Umeå University and Västerbotten County Council (ALF).

**Competing interests:** The authors have declared that no competing interests exist.

helping our body to avoid harmful stimuli, the benefits of chronic pain are more uncertain [3]. The increasing prevalence of chronic pain conditions [4] and their physical, psychological and socioeconomic consequences can have a profound impact on performance of activities in daily life and affect the individuals way of thinking, feeling and doing [5]. This have stimulated interest in underlying neurobiological mechanisms and the effects of pain on processes such as cognition. Chronic pain is associated with dysfunction in several cognitive domains [6] including mental processing speed [7], attention [8], working and episodic memory [8–10], and executive functions [6,11]. Cognitive dysfunction is now viewed as a part of the pain experience, not solely secondary to aspects such as depression or sleep difficulties [12].

The most complex cognitive functions are the executive functions, which comprise a range of complex cognitive processes such as organization, goal-directed behaviour, initiation of activity, and assessing consequences of behaviour [13]. One well-known framework for understanding executive functions is the Miyake three key cognitive components model: shifting (switching flexibly between mental sets), updating (continuous monitoring and updating of working memory), and inhibition of prepotent responses [14]. Executive functioning serves as the basis of both self-regulation [15] and emotional regulation [16], making it crucial for academic, health and behavioural outcomes [17].

Meta-analyses and systematic reviews regarding executive function in patients with chronic pain have found executive functions to be impaired within this patient group [18–20]. There are several theories about the underlying mechanisms of chronic pain and cognitive and executive functioning, including the limited cognitive resources theory [21], and the neuromodulator/neuroplasticity theory [6]. Moriarty, McGuire and Finn [6] proposed an integrated theory that incorporates all of these concepts: pain both consumes cognitive resources and affects various neuromediators as well as neural plasticity. This model also includes Corbetta and Shulman's (2002) reasoning about stimulus-driven bottom-up processes with their strong neural representation, and goal-oriented top-down processes, which both need to be balanced for effective pain modulation (Legrain et al., 2009). Imaging data supports this connection by showing that chronic pain influences networks in the prefrontal cortex, which is also an important neural structure for executive functions [22,23]. During chronic pain, changes occur in the prefrontal cortex, including alterations in neurotransmitters, glial cells, gene expression, and neuroinflammation. These changes lead to modifications in the activity, structure, and connectivity of the brain [22,23].

Executive functioning is crucial for managing all kinds of challenges in everyday life [13], which theoretically should include self-management of pain [24]. Examining the effects of executive dysfunction on pain management and interference in everyday life for these patients could thus be of high interest. However, this has not been extensively investigated in previous studies.

The aim of this exploratory study was to examine executive functioning in a large cohort of community-dwelling patients with chronic pain. Specifically, the primary objective was to investigate the association between executive functioning and pain interference in everyday life, with the hypothesis that lower executive functioning would be associated with higher degree of pain interference in everyday life. A secondary objective was to examine if executive functions in these patients were related to pain intensity, pain duration, and pain spreading, with the hypothesis that higher degree of pain intensity, duration and spreading would be negatively associated with executive functioning.

## Materials and methods

All data were collected from a large clinical trial (ClinicalTrials.gov NCT05452915). The procedure and instruments of this trial have been previously published [25]. This study complied

with all relevant national regulations, institutional policies and is in accordance with the tenets of the Helsinki Declaration (as amended in 2013) and has been approved in writing by the Swedish Ethical Review Authority (Dnr:s 2018/424-31; 2018/1235–32; 2018/2395–32; 2019–06148; 2022-02838-02). All participants received oral and written information about the study and signed informed consent before entering the study.

## Participants

All patients referred for evaluation before entering rehabilitations programs at the specialist clinics Unit of Pain Rehabilitation, Department of Rehabilitation Medicine at Danderyd University Hospital, Stockholm, Sweden (7 September 2018 to 19 October 2022) and the Department of Pain Rehabilitation, Pain Centre at Umeå University Hospital, Umeå, Sweden (1 January 2020 to 31 December 2021) were invited to participate in this study. The main inclusion criterion was chronic pain, defined as pain debut >3 months earlier. Both male and female participants were included, and inclusion age ranged from 18 to 50 years. The upper age limit was set due to interest in the trial of examining neuroinflammatory markers and functional and structural brain imaging (not reported here). Exclusion criteria were acquired brain injury (including concussion), any neurological condition, ongoing or history of severe psychiatric disorder (such as psychotic disorders, bipolar disorder, and severe depression) or substance abuse, intellectual disability, pregnancy, being unable to speak Swedish, and medical treatments that could affect cognitive functions (such as sedatives and opioids).

A total of 556 patients were screened for eligibility. Of these, 184 declined participation, 14 were unreachable, and another 158 fulfilled one or more exclusion criteria (mainly concussion or sedative medication). In addition, 11 patients did not volunteer to be a part of the Swedish Quality Registry for Pain Rehabilitation (SQRP). Thus, a total of 189 patients, 160 women and 29 men, were included in this study. Twenty of these patients did not have complete data from The Tampa Scale for Kinesiophobia, and one did not have complete data from the Multidimensional Pain Inventory. These patients were thus excluded in respective analyses containing these measures.

## Instruments and procedure

**Demographic data and self-assessment scales: The Swedish Quality Registry for Pain Rehabilitation.** Background data and results from self-assessment scales were obtained from the Swedish Quality Registry for Pain Rehabilitation (SQRP: http://www.ucr.uu.se/nrs). The SQRP contains patient data from most of the specialist pain rehabilitation centres in Sweden, covering numerous aspects of pain and other health-related variables.

Pain interference and self-management of pain were measured using three self-assessment scales from the SQRP. These scales were chosen since they are measures from the SQRP that best capture how pain interferes with everyday life. The Tampa Scale for Kinesiophobia (TSK [26]) is a 17-item self-assessment scale aiming to measure fear of movement (kinesiophobia). Each item is rated on a 1–4 Likert scale, with a higher number indicating a higher degree of kinesiophobia. The Swedish version of the TSK have been psychometrically tested and demonstrated a high degree of reliability ($\Omega = 0.82$) and discriminant validity (correlations less than 0.70) [27]. This scale was included since fear of movement leads to avoidance behaviours, that affect both ability to take part in everyday life and rehabilitation [28]. In addition two scales from the Multidimensional Pain Inventory (MPI [29]) were included as outcomes: *Interference* and *Life Control*. The items are responded to on a 7-point Likert scale, and the total sum for items are divided by the number of items in that scale, resulting in a mean score. For *Interference*, a higher number indicates more pain interference in everyday life and lifestyle.

Conversely, in *Life Control*, a higher score reflects a higher ability to manage everyday life and pain. The MPI have been validated and shown a high degree of reliability (Cronbach's α = 0.66–0.86) in the Swedish version, including a confirmatory factor analysis for the *Interference* (factor loadings 0.63–0.84) and *Life Control* (factor loadings 0.41–0.69) scales [30].

Different characteristics of pain were assessed using the following instruments: *Pain intensity* was measured by asking the patients to rate their pain intensity during the past week on the Numeric Rating Scale ranging from 0 (no pain at all) to 10 (worst possible pain). *Pain duration* was determined by asking patients to state when their pain had debuted and calculating the number of days from that date to the day they completed the SQRP questionnaires. Number of days was divided by 365 to present pain duration in years. *Pain spreading*, or number of pain locations, was obtained by asking patients to register the pain they experienced in 36 predefined anatomical areas covering the front and back of the body, on both the left side of the body (n = 18) and the right side (n = 18): head/face, neck, shoulder, upper arm, elbow, forearm, hand, anterior aspect of chest, lateral aspect of chest, belly, sexual organs, upper back, lower back, hip/gluteal area, thigh, knee, shank, and foot.

Anxiety and depression were measured by using the Hospital Anxiety and Depression Scale (HADS; [31]). The HADS is a 14-item scale, with two 7-item subscales measuring levels of anxiety and depression respectively.

The patients completed the SQRP questionnaires within one month before assessment at the clinics. At the time of the neuropsychological examination, participants were asked about their highest level of education and active medical treatment adherence.

**Neuropsychological tests.** Neuropsychological tests were administered in a fixed order by an experienced psychologist (NB or AH). Only relevant tests for this paper are described here, detailed information on the measures can be obtained from the study protocol [25]. The neuropsychological tests included in this paper represent different aspects of executive functions. Digit Span task from the fourth edition of the Wechsler Adult Intelligence Scale (WAIS-IV; [32]), is a test of auditory attention and working memory. This task is administered in two conditions: Digit Span Forwards, which comprises simple repetition of strings of digits, and Digit Span Backwards, which comprises repeating strings of digits in reversed order. Digit Span Backwards was the variable of interest in the present study since it captures working memory and thus is considered to measure an executive function. The Color-Word Interference Test (CWIT) from the Delis-Kaplan Executive Functions System (D-KEFS; [33]) primarily measures the executive functions of inhibition and switching in the conditions CWIT–Inhibition and CWIT–Switching, respectively; and the Verbal Fluency test from D-KEFS assesses executive functions, primarily word fluency and switching. This test is performed in three conditions: Word Fluency (number of words produced starting with letter F, A, and S during one minute per letter), Semantic Fluency (Number of words produced within categories animals and boys' names during one minute per category) and Semantic Shifting (number of words produced during one minute switching between the categories fruits and furniture). For all tests, higher scores indicate better performance, except for CWIT, where the score is time to completion. Raw scores were used in all analyses, to avoid influence of different normative groups in the original tests. In the presentation of the results from executive tests, a scaled score was used to visualize level of performance of the patient group as compared to normative data from the test manuals [32,33]. In these scores, a value of 10 indicates a mean score in the normative group, and the standard deviation is 3. All neuropsychological tests included show a high degree of test-retest reliability (0.75–0.82) [32,33].

All neuropsychological examinations were performed on weekdays between 10 AM and 2 PM and lasted about one hour. For additional details regarding instruments and procedure, see the published study protocol [25].

## Statistical analyses

The data analysis was performed in three stages. Stage one was to identify and report demographic and descriptive data from self-assessment scales and neuropsychological tests. In stage two, bivariate correlations (Pearson) were used to examine relationships between performance on executive tests within the patient group and self-assessment scales of pain interference in everyday life. Multiple linear regressions were also performed regarding these analyses using relevant covariates, to establish that relationships between performance on executive tests and self-assessment scales of pain interference in everyday life were not due to any of these factors. Age, gender, years of education, and self-assessment of depression were included as covariates due to their established relationship to both pain [34] and executive functions [35]. Since anxiety and depression score of the HADS in line with previous studies [36] were highly correlated (r = 0.53, p < 0.001), only depression was included as a covariate in the multiple regression analyses to avoid the problem of multicollinearity. Similarly, in the third and final stage, bivariate correlations (Pearson) were used to examine relationships between pain characteristics and performance on executive tests within the patient group. These correlations were analysed further using multiple linear regression and relevant covariates.

The normality assumption was tested in all stages through visual inspection of histograms and Q-Q plots. This showed a normal distribution of all variables, warranting the use of parametric tests in stages two and three. Visual inspection of histograms for the regression standardized residual showed normality of residuals for the linear regressions in stages two and three.

All analyses were conducted using version 28.0 of SPSS Statistics (IBM Corporation, Armonk, New York), with a significance level set to $p < .05$.

# Results

## Patient characteristics

Demographic data and results from self-assessment scales for patients with chronic pain are presented in Table 1. A majority (85%) of the patients were women, and the mean age was 33.15 (SD 8.51). The mean number of years of education was 13.13 (SD 2.26), which is above the number of years in Swedish upper secondary school, indicating a high level of education in the sample. 61% of the patients scored above the suggested cutoff of 8 [31] on the HAD Anxiety scale, and 57% scored above the cutoff on the HADS Depression scale.

## Executive functioning, kinesiophobia, and pain interference

In the unadjusted analyses, several results from tests of executive functions were significantly correlated with pain interference in everyday life (Table 2). All executive measurements, except for Digit Span, were significantly correlated to kinesiophobia. All significant correlations were in the same direction, with a higher performance on the executive test being correlated with lower self-rated kinesiophobia. For pain interference, better performance on Semantic Fluency, Switching Fluency, and CWIT-Inhibition was correlated with lower self-assessed interference of pain in everyday life. Additionally, for self-rated life control, better performance on tests of Semantic fluency and Semantic Switching was correlated with better self-rated life control.

Using multiple linear regression and adjusting for relevant covariates made four of these correlations no longer significant (Table 3). Six of the correlations (Fig 1) were still significant after adjustment for age, gender, years of education, and HADS Depression. In addition, the

**Table 1. Demographic and baseline characteristics.**

|  | All patients (n = 189) |  |
|---|---|---|
| **Female Gender** | 160 (85%) |  |
| **Age** | 33.15 (8.51) |  |
| **Years of education** | 13.13 (2.26) |  |
| **Pain Characteristics** |  |  |
| • Pain Duration (Years since pain debut) | 8.95 (7.15) |  |
| • Pain Spreading (Number of Pain Localizations) | 16.32 (8.09) |  |
| • Pain Intensity (Last week) | 6.45 (1.61) |  |
| **Self-assessment Scales** |  |  |
| • HADS Anxiety | 9.28 (4.55) |  |
| • HADS Depression | 8.34 (4.04) |  |
| • TAMPA Scale of Kinesiophobia | 37.50 (8.08) |  |
| • MPI Interference | 4.06 (1.10) |  |
| • MPI Life Control | 2.65 (1.09) |  |
| **Neuropsychological test data** | **Raw Scores** | **Scaled Score** |
| • Digit Span Backwards | 7.74 (1.98) | 8 |
| • Word Fluency | 39.79 (11.58) | 10 |
| • Semantic Fluency | 44.67 (9.27) | 12 |
| • Switching Fluency | 14.90 (2.87) | 10 |
| • CWIT - Inhibition | 53.79 (15.08) | 10 |
| • CWIT - Switching | 65.57 (19.80) | 9 |

Note: Data presented are Mean (Standard Deviation) or count (percentages). MPI = Multidimensional Pain Inventory; CWIT = Color-Word Interference Test; Scaled scores were obtained from normative data within the test manuals Wechsler (2008) and Delis et al (2001), where 10 is the mean score and 3 is the standard deviation in the normative population.

relationship between Word Fluency and MPI Interference reached significance after this adjustment.

Several other factors in the multiple regression models also showed a significant relationship to the self-assessment of pain interference in everyday life, especially HADS depression score, which was a significant contributor in all models. Education, age, and gender was a significant contributor in all models with TSK as the dependent variable, but not in any other models. The results from all multiple regressions with scales of self-assessment of pain interference in everyday life as dependent variables are presented (S1-S18 Tables in S1 File).

## Executive functioning and pain characteristics

The relationships between the pain characteristics pain intensity, pain duration, and pain spreading (number of pain localizations) and executive test results within the chronic pain patient group were first investigated in an unadjusted correlation analysis (Table 4). Word Fluency was negatively correlated to pain intensity, meaning that higher pain was associated with lower production of words. CWIT–Inhibition was positively correlated in both cases, indicating worse performance with higher pain intensity. Pain duration was positively correlated to Digit Span Backwards, Word Fluency, and Semantic Fluency, indicating that those with longer pain duration had better performance on executive tasks. Number of pain localizations was not significantly correlated to any executive function measure.

**Table 2. Results from linear regression with executive functions as predictors and self assessment scales as outcomes.**

|  | TAMPA Scale of Kinesiophobia | | MPI Interference | | MPI Life Control | |
|---|---|---|---|---|---|---|
|  | r | p | r | p | r | p |
| Digit Span Backwards | -0.06 | 0.45 | -0.05 | 0.51 | -0.03 | 0.71 |
| Word Fluency | -0.18 | **0.02** | -0.14 | 0.05 | 0.12 | 0.10 |
| Semantic Fluency | -0.22 | **<0.01** | -0.24 | **<0.01** | 0.25 | **<0.001** |
| Switching Fluency | -0.20 | **<0.01** | -0.21 | **<0.01** | 0.24 | **<0.001** |
| CWIT - Inhibition | 0.21 | **<0.01** | 0.18 | **0.01** | -0.13 | 0.08 |
| CWIT - Switching | 0.27 | **<0.001** | 0.12 | 0.10 | -0.08 | 0.26 |

Note: CWIT = Color-Word Interference Test; MPI = Multidimensional Pain Inventory.

Multiple linear regressions were performed to adjust for covariates: age, gender, years of education, and HADS Depression. After doing so, all relationships between pain characteristics and tests of executive function were no longer significant (all $ps > 0.05$).

## Discussion

The aim of the present study was to investigate the relationship between executive function and pain interference in everyday life, as well as the relationship to different pain characteristics in patients with chronic pain. We found that both switching and inhibition, but not updating, were associated with pain interference and life control, even when adjusting for age, depression, and education. This finding adds to previous research showing that deficits in executive functions are not only present in the chronic pain population [18–20], but also might affect the everyday life of these patients and might influence rehabilitation outcomes.

Verbal fluency, including switching, was associated with both pain interference and life control in patients with chronic pain. Worse ability to produce semantic words in a rapid and flexible manner in the Semantic and Switching Fluency tests were all related to a higher degree of pain interference and lower life control. These fluency tests are broad measures of executive functioning, reflecting both self-initiation and strategy generation [33,37], as well as switching [33,37,38]. Thus, better fluency indicates an ability to generate new ideas and problem-solve, and being flexible reflects the ability to adapt your actions in relation to new demands. Both these functions could make it easier for patients who are suffering from chronic pain to explore new thoughts and behaviours and adjust their lives accordingly.

Inhibition, which is the ability to inhibit overlearned impulses in a flexible way, was significantly associated with the kinesiophobia, where worse performance on the inhibition test was

**Table 3. Results from multiple linear regressions with executive functions as predictors and self assessment scales as outcomes.**

|  | TAMPA Scale of Kinesiophobia (n = 169) | | | MPI Interference (n = 188) | | | MPI Life Control (n = 188) | | |
|---|---|---|---|---|---|---|---|---|---|
|  | B | Std. Error | p | B | Std. Error | p | B | Std. Error | p |
| Digit Span Backwards | -0.213 | 0.30 | .478 | -0.04 | 0.04 | .311 | -0.01 | 0.04 | .878 |
| Word Fluency | -0.07 | 0.05 | .152 | -0.01 | 0.01 | **.045** | 0.01 | 0.01 | .134 |
| Semantic Fluency | -0.09 | 0.07 | .178 | -0.02 | 0.01 | **.020** | 0.16 | 0.01 | **.044** |
| Switching Fluency | -0.24 | 0.21 | .265 | -0.06 | 0.03 | **.035** | 0.06 | 0.03 | **.015** |
| CWIT - Inhibition | 0.09 | 0.04 | **.026** | 0.01 | 0.01 | .171 | -0.00 | 0.01 | .538 |
| CWIT - Switching | 0.10 | 0.03 | **.002** | 0.01 | 0.00 | .152 | -0.00 | 0.00 | .364 |

Note: CWIT = Color-Word Interference Test; MPI = Multidimensional Pain Inventory; Covariates: Age, Gender, Years of Education, HADS Depression.

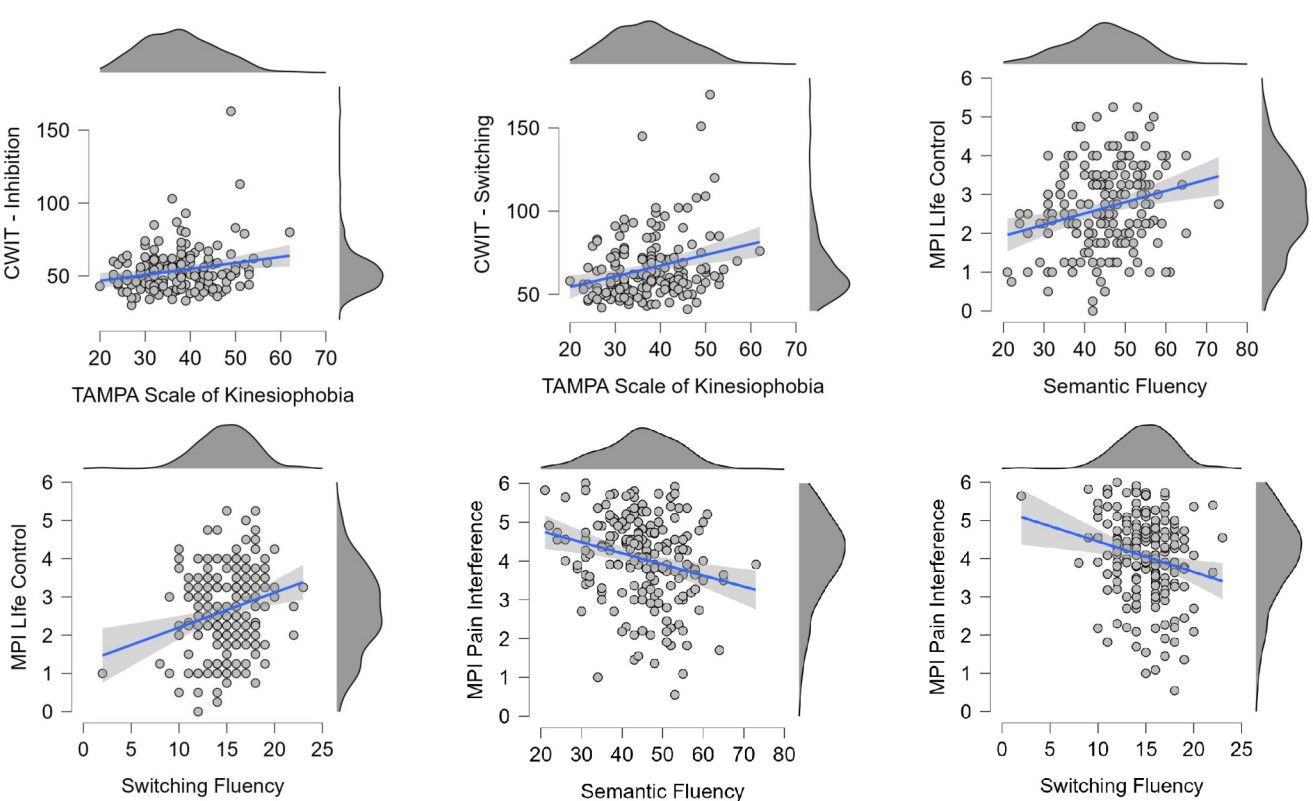

**Fig 1. Scatterplots of results from executive tests and self-assessment of kinesiophobia, pain interference and life control.** Note: CWIT = Color-Word Interference Test; MPI = Multidimensional Pain Inventory.

related to a higher degree of kinesiophobia. The TSK is supposed to measure fear of movement [26], which is a common fear in patients with chronic pain [27]. This fear predicts outcome of physical disability and pain severity for these patients [39]. Reducing fear of movement and encouraging patients with chronic pain to get physically active is a key aspect of many pain rehabilitation programs, since physical activity could both reduce pain and the fear of movement itself [40]. Results on the TSK have also been associated with catastrophic thinking [41]. Behavioral inhibition is a key aspect of many kinds of anxiety and worrying, including pain catastrophizing [42], as the ability to inhibit worrying thoughts is a prerequisite for not getting

**Table 4. Bivariate Correlations between pain charateristics and executive functions in patients.**

|  | Pain Intensity | | Pain Duration | | Number of Pain Localizations | |
|---|---|---|---|---|---|---|
|  | r | p | r | p | r | p |
| **Digit Span Backwards** | -0.09 | 0.24 | 0.17 | **0.03** | 0.03 | 0.65 |
| **Word Fluency** | -0.16 | **0.03** | 0.19 | **0.01** | 0.01 | 0.90 |
| **Semantic Fluency** | -0.09 | 0.26 | 0.20 | **0.01** | 0.00 | 0.97 |
| **Switching Fluency** | -0.05 | 0.49 | -0.004 | 0.96 | 0.08 | 0.29 |
| **CWIT - Inhibition** | 0.16 | **0.03** | -0.07 | 0.38 | 0.08 | 0.27 |
| **CWIT - Switching** | 0.14 | 0.06 | -0.09 | 0.25 | 0.07 | 0.34 |

Note: CWIT = Color-Word Interference Test.

stuck in excessive fear. Thus, the executive function inhibition could theoretically be linked to the ability to inhibit the fear of movement, a theory empirically strengthened by our data.

The executive function updating was measured by the Digit Span Backwards task in the present study. Performance on this working memory task was not related to pain characteristics, nor to pain interference. This latter aspect is in line with one previous study [43] investigating the relationships between executive functions and pain disability. While Elkana, Conti [43] suggested that this lack of association between working memory and pain disability might be due to the short duration of the task, there might be other explanations. Brain imaging studies have shown working memory manipulation tasks, such as the Digit Span Backwards, to be associated to activity in dorsofrontal/frontoparietal networks, while updating in addition also rely on striato-cortical networks [44]. Working memory is still considered an executive function [45], but the intended updating function may be better investigated using other tasks, such as the Keep track, Tone Monitoring or Letter Memory task as used in Miyake and colleagues' original study [14]. Thus, future studies could use more specific updating tasks to investigate whether updating is associated with pain characteristics or pain interference in everyday life.

The regression analyses revealed that pain spreading, pain duration, and pain intensity were not reliably related to executive functions. This is consistent with previous research. Several studies using the same executive measures as in the present study, have not been able to establish any relationships between pain intensity and the Digit Span task [46–50], the Stroop test [46,50–53], or verbal fluency tests [49,50,52,53]. Oosterman, Derksen [51] as well as Kurita, de Mattos Pimenta [47] failed to find relationships between pain duration and executive functions measured with the Trail Making Test–B, Stroop test, Digit Span and the Zoo Map test from the Behavioural Assessment of the Dysexecutive Syndrome [54]. Further, the analyses regarding associations between spreading of pain (number of pain localizations) and executive functioning revealed no significant associations. This relationship has not been very well studied; to our knowledge, only one study has previously examined this. Kurita, de Mattos Pimenta [47] investigated the relationship between several cognitive tests and pain characteristics in 49 patients with chronic pain. Neither basic psychomotor functions, such as reaction time and finger tapping, nor executive functions, captured with the Trail Making Test–B and Digit Span Backwards, were significantly related to pain duration, pain intensity, or spreading of pain (number of pain localizations). Our large-scale study confirms these findings, and it seems that pain intensity, pain duration, and pain spreading are not related to the aspects of the Miyake inhibition, switching, and updating model of executive functions, at least not when studied using Digit Span Backwards, the Stroop test, and verbal fluency tests.

Previous studies have often viewed depression as a symptom that often exacerbates symptoms of pain and pain management, leading some authors to label terms such as the "depression-pain syndrome" [55] or "depression-pain dyad" [56]. The results of the present study do indeed show that depression is associated with outcomes regarding pain interference and management (see supplemental data). Thus, depression seems to be an important contributor, but not the only one. After adjusting for the effects of depression, executive functioning was still associated with these outcomes. A recent study showed similar results, indicating that shifting and inhibition, but not updating, are related to pain disability, regardless of depression [43]. In addition, a systematic review investigating executive functions in patients with chronic pain, found that depression only seems to play a minor role in explaining the association between chronic pain and executive functioning [20]. In conclusion, while depression is common in patients with chronic pain, it cannot alone explain deficits in executive functioning, and executive functioning in itself might affect how pain interferes with everyday life and pain management, regardless of whether depression is present or not.

## Clinical implications

The direction of causality must always be discussed in observational correlation studies. Executive functioning, especially cognitive flexibility, can actually predict which patients will develop long-lasting or chronic pain [57]. Thus, perhaps it is not the pain that causes executive dysfunction, but rather the executive dysfunction that affects the development of chronic pain. Either way, the executive difficulties that affect everyday pain interference seem to be present in this patient cohort and might need to be addressed both in rehabilitation and in future studies. Our study indicates that executive functions need to be considered for patients who are candidates for pain rehabilitation. Evaluating patients with chronic pain regarding inhibition may help in understanding who is at risk of developing fear of movement, and to guide the rehabilitation for these patients. By assessing verbal fluency and switching patients that have difficulties in finding new strategies for coping with pain may be identified. However, the rehabilitation needs to go further and offer strategies to enhance the executive functions in question. For acquired brain injuries there is evidence of functional improvement with executive training [58]. Studies on the effects of cognitive/neuropsychological rehabilitation in chronic pain are still sparse. Therefore, future studies should be conducted to evaluate the effect of cognitive training on coping with chronic pain.

## Limitations and future studies

Since this study has a correlational design, the causal relationship between executive functioning and pain interference in everyday life cannot be determined solely on these results, and thus warrants further investigation in future studies with other designs and better experimental control. The study included a large sample of chronic pain patients, a major limitation was the rather heterogeneous patient group; although they all experienced chronic pain, and most patients suffered from musculoskeletal chronic pain, this pain varied in type and aetiology. The focus of this study was however solely to examine the characteristics of pain and pain interference and their relation to executive function, but forthcoming studies need to investigate differences between different pain diagnoses regarding these relationships. In addition, the sample was rather young in comparison to the majority of patients with chronic pain. The results may thus not be generalizable to the general population of patients with chronic pain, but rather to younger groups within this population. Furthermore, we mainly used timed executive tasks in this study. Mental speed seems to be more reliably affected in patients with chronic pain [6,59,60], and hence might be a mediator or moderator in the relationship between chronic pain and executive functions. Future studies should also include executive tasks that do not rely on time. In addition, other aspects within the complex umbrella of executive functions might be more affected by chronic pain. It has even been argued that more complex executive-type functions might be more strongly affected than less complex automated tasks [6]. Thus, future studies should incorporate other, and possibly more complex tasks, when examining executive functions in patients with chronic pain.

## Conclusions

Impaired executive functions appear to be common in patients with chronic pain. However, there does not seem to be a link between executive functioning and pain characteristics such as pain intensity, pain duration, or pain spreading. Yet, executive functioning seems associated with how much pain interferes in the everyday life of patients with chronic pain. Thus, a focus on evaluating executive functions (e.g., through neuropsychological assessment) and their remediation may help in tailoring the rehabilitation and management of pain in everyday life.

## Supporting information

**S1 File. Multiple regression analyses for TAMPA scale of kinesiophobia, MPI interference, and MPI life control.**
(DOCX)

## Acknowledgments

The authors would like to thank Karin Wermelin and late Nina von Rüdiger for initial data preparation for parts of this study.

## Author Contributions

**Conceptualization:** Nils Berginström, Sofia Wåhlin, Linn Österlund, Anna Holmqvist, Monika Löfgren, Britt-Marie Stålnacke, Marika C. Möller.

**Data curation:** Nils Berginström, Sofia Wåhlin, Linn Österlund, Anna Holmqvist, Marika C. Möller.

**Formal analysis:** Nils Berginström, Sofia Wåhlin, Linn Österlund, Anna Holmqvist, Marika C. Möller.

**Funding acquisition:** Nils Berginström, Monika Löfgren, Britt-Marie Stålnacke Marika C. Möller.

**Methodology:** Nils Berginström, Monika Löfgren, Britt-Marie Stålnacke, Marika C. Möller.

**Project administration:** Nils Berginström, Anna Holmqvist, Marika C. Möller.

**Supervision:** Marika C. Möller.

**Visualization:** Nils Berginström.

**Writing – original draft:** Nils Berginström, Sofia Wåhlin, Linn Österlund.

**Writing – review & editing:** Nils Berginström, Sofia Wåhlin, Linn Österlund, Anna Holmqvist, Monika Löfgren, Britt-Marie Stålnacke, Marika C. Möller.

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
