## [Decision Letter · Decision Letter 0]

11 Aug 2024

PONE-D-24-24703Executive Functioning is Associated to Everyday Interference of Pain in Patients with Chronic PainPLOS ONE

Dear Dr. Berginström,

Thank you for submitting your manuscript to PLOS ONE. After careful consideration, we feel that it has merit but does not fully meet PLOS ONE’s publication criteria as it currently stands. Therefore, we invite you to submit a revised version of the manuscript that addresses the points raised during the review process.

We look forward to receiving your revised manuscript.

Kind regards,

Vincenzo De Luca

Academic Editor

PLOS ONE

Journal Requirements:

2. Thank you for stating the following financial disclosure: "This study was funded by Karolinska Institutet, Department of Clinical Sciences, the Promobilia Foundation (no. A22056), by the research and development fund granted by Västerbotten County Council, and through a regional agreement between Umeå University and Västerbotten County Council (ALF)."  

5. We note you have included a table to which you do not refer in the text of your manuscript. Please ensure that you refer to Table 4 in your text; if accepted, production will need this reference to link the reader to the Table.

Reviewers' comments:

Reviewer's Responses to Questions

**Comments to the Author**

1. Is the manuscript technically sound, and do the data support the conclusions?

Reviewer #1: No

Reviewer #2: Partly

2. Has the statistical analysis been performed appropriately and rigorously? 

Reviewer #1: Yes

Reviewer #2: N/A

3. Have the authors made all data underlying the findings in their manuscript fully available?

Reviewer #1: Yes

Reviewer #2: Yes

4. Is the manuscript presented in an intelligible fashion and written in standard English?

Reviewer #1: Yes

Reviewer #2: Yes

5. Review Comments to the Author

Reviewer #1: In the present study, the authors investigated the relationship of chronic pain and executive functions. They report that lower scores in test of executive functioning were related to pain interference and a reduced life control. Pain characteristics (duration, intensity, spreading) were not related with measures of executive functioning. Additionally, they found an association with depression and pain self-management. They conclude that the assessment of executive functioning may help in the rehabilitation of patients with chronic pain.

The authors performed a rigorous data collection and report their Methods and Result in a comprehensive, straightforward matter. However, the Introduction of the manuscript would benefit from a more detailed elaboration of the research background to be able to follow the authors’ line of thought and the development of their research question. Unfortunately, the authors only report correlational evidence which is unable to support the strong conclusions they draw from their study. Nonetheless, the evidence the authors gathered is of interest for researchers and clinicians. Therefore, I suggest the authors to revise the manuscript to tune down their interpretations and stress more clearly the weaknesses of correlational findings. My detailed suggestions/comments on Major and Minor points of criticism are outlined below.

----Major----

Abstract

- The authors state that “lower performance on tests of executive functioning was related to a higher degree of pain interference and a lower degree of life control due to pain” (ll. 29-30) – can it be concluded from the data that the lower degree of life control was due to the pain? Could it also be that patients that, for some reason, have lower control over their lives experience chronic pain more severely (e.g., due to generally reduced life satisfaction)?

Introduction

-The authors rapidly jump to the association between chronic pain and cognition. While this is straightforward, I would suggest to start with a more basic introduction of chronic pain, e.g. that it is not (necessarily) based on any physical problems and has no biological value. For readers who do not know this it might be confusing to jump to the cognitive side of the problem so rapidly.

-Several aspects in the introduction need more explaining. For example, the authors mention the “cognitive resources theory” (l. 66) and the “neuromodulator/neuroplasticity theory” (l. 67) but do not present any further information about these theories. It would be important to know more details about these theories to be able to follow the authors’ line of thought on why the current study is necessary to understand the relationship between chronic pain and executive functions.

-Why is this study framed as an “exploratory study” (l. 73)? The authors present relevant literature that justifies the investigation of chronic pain and cognitive functions quite well.

-Related to the previous comment, the authors should formulate their hypotheses more clearly at the end of the introduction.

Materials and Methods

-Did the authors exclude participants who reported a history of drug abuse? Aside from “medical treatment” (l. 103) this could also affect both cognitive functions as well as pain processing.

-Why did the authors decide to use “raw scores” (l. 176) for the analyses instead of standardized scores?

-Why did the authors not use “gender” or “sex” as a covariate in their analyses? It is widely known that gender/sex influences pain perception.

-The authors should elaborate on why they included measures of kinesiophobia (l. 123)

-In the introduction, the authors rightly criticize that the relationship between chronic pain and executive functions has been addressed using correlational evidence. However, they also report merely correlational evidence themselves. Therefore, I am unsure about the merit of this work, especially as a standalone manuscript. Certainly, the authors conducted a rigorous study, but the work presented here would rather be a side note on e.g. data on “neuroinflammatory markers and functional and structural brain imaging” (l.99 -100) they refer to, but do not report here.

-Did the authors correct for multiple comparisons? This would be appropriate when performing multiple correlations.

Discussion

-The authors make strong interpretations based on their correlational analyses and combine them with very broad speculations based on other findings reported in literature. These conclusions the authors draw are not justified based on their study, especially since the lack of an experimental manipulation does not allow for any interpretations of causality.

----Minor----

Abstract

-what does “everyday interference of pain” (l. 27) refer to? Is this how much pain interferes with the everyday life of the participants?

- The sentence “Depression was also associated with self management of pain, indicating that patients with lower executive functioning in combination with depression may need special attention in the rehabilitation of patients with chronic pain” (ll. 34-36) seems a bit odd; I suggest shortening to “Depression was also associated with self management of pain, indicating that patients with lower executive functioning in combination with depression may need special attention during rehabilitation”

Materials and Methods

-The authors state that “Both male and female participants were included” (l. 97) – did they specifically exclude people with other gender identities?

-Why were not all patients included in the linear regression analyses? Additionally, why was the sample size lower for the TAMPA Scale of Kinesiophobia regression?

Discussion

-Sudden change of font color on page 20

-In their conclusion, the authors write “Affected executive functions seem to be common in patients with chronic pain. Pain characteristics such as pain intensity, pain duration, and pain spreading do not seem to be associated with executive functioning.” These statements seem like direct contradictions of each other, which is a bit confusing. I suggest rephrasing.

Reviewer #2: The reviewer enjoyed reading the article in general. Some general suggestions from the reviewer might enhance the reading experience, such as, clearer sentence structure and consistent referencing style.

The introduction section gave a brief picture about the relationship between executive function and pain. It also provided a brief background of the current study, but would appreciate it if the authors could extend on the theoretical assumptions of the study with explicitly stated hypotheses of the directions of impacts. In addition, the authors might need to rephrase on ref number 17 about availability of interference and experimental studies over correlational studies.

In the methodology parts, the reviewer would suggest to the authors that repeating exclusion criteria, upper age limit, and the demographic variables from ref number 20 would assist the readers to follow the argument. Statistical features of the psychometrics and external validity of both of the scales and the neuropsychological tests would also be necessary to equip readers with an enhanced understanding of the measures. Restructuring the paragraphs under "instruments and procedures" into two separate sections would also be helpful in smoothening the understanding of the methodology parts.

The reviewer would also like to suggest the inclusion of reference between lines 192-194 so that readers can better understand the established relationship between the mentioned variables with pain and executive functions (possibly they might have been discussed in the introduction). Moreover, there may be a necessity to state explicitly on the multicollinearity of the HADS subscales with previous evidence and with the current dataset so that the authors might have an option to either choose to include only one subscale or to compute them into a single measurement with better support from the evidence. Lastly the authors are suggested to take into consideration of statistical tests, e.g., the Kolmogorov-Smirnov test, instead of visual inspection of the Gaussian distribution.

Regarding the results and discussion of the paper, the reviewers would like to suggest that the conversion between raw scores and scaled scores may be difficult for readers to follow and may require some elaboration of those data, together with the effect size of the statistics to report the magnitude of the associations. It was also observed that the direction of CWIT may be different from the other measures, which further brings up the question if the authors would consider putting all dependent variables in a single model consisting of all the executive functions measures with examination of multicollinearity included. The same suggestion lies also between exploration of associations between executive functions and pain characteristics.

6. PLOS authors have the option to publish the peer review history of their article (what does this mean?). If published, this will include your full peer review and any attached files.

Reviewer #1: **Yes: **Sophie Siestrup

Reviewer #2: No

---

## [Author Response · Author response to Decision Letter 0]

20 Sep 2024

All references to lines within the manuscript is within the manuscript file with track changes. 

Journal Requirements:

Author response: The manuscript and all other files have now been adjusted to PLOS ONE’s style requirements, including file naming.

2. Thank you for stating the following financial disclosure: "This study was funded by Karolinska Institutet, Department of Clinical Sciences, the Promobilia Foundation (no. A22056), by the research and development fund granted by Västerbotten County Council, and through a regional agreement between Umeå University and Västerbotten County Council (ALF)." 

Author response: The funding statement have been changed according to the above instructions, and have been included in the cover letter: “This study was funded by Karolinska Institutet, Department of Clinical Sciences, the Promobilia Foundation (no. A22056), by the research and development fund granted by Västerbotten County Council, and through a regional agreement between Umeå University and Västerbotten County Council (ALF). The funders had no role in study design, data collection and analysis, decision to publish, or preparation of the manuscript.”

Author response: We have now discussed this within the research group and with ethical review boards advisors, and decided that an anonymized data file can be uploaded to a data repository. The data file is available at the zenodo repository: https://doi.org/10.5281/zenodo.13809000. This information has been included in the editorial manager submission system. 

Author response: Captions for supporting information have now been included in the end of the manuscript, and the in-text citations now match these captions (Line 285).

5. We note you have included a table to which you do not refer in the text of your manuscript. Please ensure that you refer to Table 4 in your text; if accepted, production will need this reference to link the reader to the Table.

Author response: A reference to table 4 have now been included in the main text (Line 290-291)

Reviewer #1: 

In the present study, the authors investigated the relationship of chronic pain and executive functions. They report that lower scores in test of executive functioning were related to pain interference and a reduced life control. Pain characteristics (duration, intensity, spreading) were not related with measures of executive functioning. Additionally, they found an association with depression and pain self-management. They conclude that the assessment of executive functioning may help in the rehabilitation of patients with chronic pain.

The authors performed a rigorous data collection and report their Methods and Result in a comprehensive, straightforward matter. 

Author response: We thank the reviewers for these kind remarks about our paper. 

However, the Introduction of the manuscript would benefit from a more detailed elaboration of the research background to be able to follow the authors’ line of thought and the development of their research question. Unfortunately, the authors only report correlational evidence which is unable to support the strong conclusions they draw from their study. Nonetheless, the evidence the authors gathered is of interest for researchers and clinicians. Therefore, I suggest the authors to revise the manuscript to tune down their interpretations and stress more clearly the weaknesses of correlational findings. 

Author response: The reviewer makes a very important mark about the conclusions and implications of our study, and we agree that the manuscript was in many parts too strongly formulated. We believe that by following the suggestions of the reviewer below, we have revised the manuscript in such a manner, by toning down some interpretations, and included a limitations section on the evidence value of our correlational study. 

My detailed suggestions/comments on Major and Minor points of criticism are outlined below.

----Major----

Abstract

- The authors state that “lower performance on tests of executive functioning was related to a higher degree of pain interference and a lower degree of life control due to pain” (ll. 29-30) – can it be concluded from the data that the lower degree of life control was due to the pain? Could it also be that patients that, for some reason, have lower control over their lives experience chronic pain more severely (e.g., due to generally reduced life satisfaction)?

Author response: We agree with the reviewer that these conclusions were not very well formulated, and have thus changed the phrasing of this paragraph: “This indicates that lower performance on tests of executive functioning was significantly associated with a higher degree of pain interference and a lower degree of life control.” (Line 30-32)

Introduction

-The authors rapidly jump to the association between chronic pain and cognition. While this is straightforward, I would suggest to start with a more basic introduction of chronic pain, e.g. that it is not (necessarily) based on any physical problems and has no biological value. For readers who do not know this it might be confusing to jump to the cognitive side of the problem so rapidly.

Author response: We thank the reviewer for this suggestion, and have changed the first part of the introduction accordingly: “Chronic pain, defined as pain persisting or recurring for longer than 3 months, is associated with significant emotional distress and functional disability [1]. The new definition of pain from the International Association for the Study of Pain (2) states pain to be an unpleasant sensory or emotional experience that can be, but not necessarily is, associated with actual tissue damage. The new definition and notes were revised to better explain the complexity of pain in order to better assess and manage people with pain. While acute pain plays a protective role by helping our body to avoid harmful stimuli, the benefits of chronic pain are more uncertain [3].The increasing prevalence of chronic pain conditions [4] and their physical, psychological and socioeconomic consequences can have a profound impact on performance of activities in daily life and affect the individuals way of thinking, feeling and doing [5]” (Line 47-57). 

-Several aspects in the introduction need more explaining. For example, the authors mention the “cognitive resources theory” (l. 66) and the “neuromodulator/neuroplasticity theory” (l. 67) but do not present any further information about these theories. It would be important to know more details about these theories to be able to follow the authors’ line of thought on why the current study is necessary to understand the relationship between chronic pain and executive functions.

Author response: We thank the reviewer for suggesting this, and have now included a section on the theories behind cognitive and executive dysfunction in patients with chronic pain: “There are several theories about the underlying mechanisms of chronic pain and cognitive and executive functioning, including the limited cognitive resources theory (4), and the neuromodulator/neuroplasticity theory (5). Moriarty, McGuire and Finn (5) proposed an integrated theory that incorporates all of these concepts: pain both consumes cognitive resources and affects various neuromediators as well as neural plasticity. This model also includes Corbetta and Shulman’s (2002) reasoning about stimulus-driven bottom-up processes with their strong neural representation, and goal-oriented top-down processes, which both need to be balanced for effective pain modulation (Legrain et al., 2009). Imaging data supports this connection by showing that chronic pain influences networks in the prefrontal cortex, which is also an important neural structure for executive functions (6, 7). During chronic pain, changes occur in the prefrontal cortex, including alterations in neurotransmitters, glial cells, gene expression, and neuroinflammation. These changes lead to modifications in the activity, structure, and connectivity of the brain (6, 7).” (Line 75-88)

-Why is this study framed as an “exploratory study” (l. 73)? The authors present relevant literature that justifies the investigation of chronic pain and cognitive functions quite well.

-Related to the previous comment, the authors should formulate their hypotheses more clearly at the end of the introduction.

Author response: We thank the reviewer for this comment, and agree that the association between chronic pain and cognitive impairments are quite established. However, the association between executive functioning and everyday interference of pain have not been previously studied in chronic pain patients, and thus we frame the study exploratory. Still, we agree with the reviewer that a clearly stated hypothesis would be appropriate. The objective section now reads: ”The aim of this exploratory study was to examine executive functioning in a large cohort of community-dwelling patients with chronic pain. Specifically, the primary objective was to investigate the association between executive functioning and pain interference in everyday life, with the hypothesis that lower executive functioning would be associated with higher degree of pain interference in everyday life. A secondary objective was to examine if executive functions in these patients were related to pain intensity, pain duration, and pain spreading, with the hypothesis that higher degree of pain intensity, duration and spreading would be negatively associated with executive functioning.” (Line 96-104) 

Materials and Methods

-Did the authors exclude participants who reported a history of drug abuse? Aside from “medical treatment” (l. 103) this could also affect both cognitive functions as well as pain processing.

Author response: Yes, these participants were excluded, following the study protocol. This has now been clarified in the test: “Exclusion criteria were acquired brain injury (including concussion), any neurological condition, ongoing or history of severe psychiatric disorder (such as psychotic disorders, bipolar disorder, and severe depression) or substance abuse,…” (Line 126-129)

-Why did the authors decide to use “raw scores” (l. 176) for the analyses instead of standardized scores?

Author response: This was done to avoid influences of different normative groups in the tests, which have now been clarified in the text: “Raw scores were used in all analyses, to avoid influence of different normative groups in the original tests.” (Line 209-210)

-Why did the authors not use “gender” or “sex” as a covariate in their analyses? It is widely known that gender/sex influences pain perception.

Author response: This is a completely true statement from the reviewer, and we do agree that gender should be included as a covariate. However, as can be seen in the revised table 3, this does not change the results very much, and the conclusions drawn from the results are the same.

-The authors should elaborate on why they included measures of kinesiophobia (l. 123)

Author response: Thank you for pointing out this omission. We have now included a sentence on this matter, with a reference to a recent review: “This scale was included since fear of movement leads to avoidance behaviours, that affect both ability to take part in everyday life and rehabilitation (8).” (Line 156-158)

-In the introduction, the authors rightly criticize that the relationship between chronic pain and executive functions has been addressed using correlational evidence. However, they also report merely correlational evidence themselves. Therefore, I am unsure about the merit of this work, especially as a standalone manuscript. Certainly, the authors conducted a rigorous study, but the work presented here would rather be a side note on e.g. data on “neuroinflammatory markers and functional and structural brain imaging” (l.99 -100) they refer to, but do not report here.

Author response: We agree with the reviewer that criticizing correlational studies when we are presenting correlational data ourselves is problematic, and have thus removed this statement. However, we do believe that this manuscript has merit on its own, especially since the association between executive functioning and everyday interference of pain have not been extensively studied before. We have tried to clarify this in the introduction, before presenting the aim of the study: ”Executive functioning is crucial for managing all kinds of challenges in everyday life (12), which theoretically should include self-management of pain (23). Examining the effects of executive dysfunction on pain management and interference in everyday life for these patients could thus be of high interest. However, this has not been extensively investigated in previous studies.” (Line 91-95) 

-Did the authors correct for multiple comparisons? This would be appropriate when performing multiple correlations.

Author response: We absolutely agree with the reviewer that adjusting for multiple comparison is important when performing multiple correlation. However, in regard to our previous comment about this being an exploratory study, we did not want to make type-II-errors and missing out on reporting possibly interesting results. We also added a short note on this in the limitations section regarding future studies:

---

## [Decision Letter · Decision Letter 1]

21 Oct 2024

Executive Functioning is Associated to Everyday Interference of Pain in Patients with Chronic Pain

PONE-D-24-24703R1

Dear Dr. Berginström,

We’re pleased to inform you that your manuscript has been judged scientifically suitable for publication and will be formally accepted for publication once it meets all outstanding technical requirements.

Kind regards,

Vincenzo De Luca

Academic Editor

PLOS ONE

Additional Editor Comments (optional):

Reviewers' comments:

Reviewer's Responses to Questions

**Comments to the Author**

1. If the authors have adequately addressed your comments raised in a previous round of review and you feel that this manuscript is now acceptable for publication, you may indicate that here to bypass the “Comments to the Author” section, enter your conflict of interest statement in the “Confidential to Editor” section, and submit your "Accept" recommendation.

Reviewer #1: All comments have been addressed

2. Is the manuscript technically sound, and do the data support the conclusions?

Reviewer #1: Yes

3. Has the statistical analysis been performed appropriately and rigorously? 

Reviewer #1: Yes

4. Have the authors made all data underlying the findings in their manuscript fully available?

Reviewer #1: Yes

5. Is the manuscript presented in an intelligible fashion and written in standard English?

Reviewer #1: Yes

6. Review Comments to the Author

Reviewer #1: I congratulate Dr Berginström and his colleagues on doing a great job on the revision. All of my points were addressed adequately and convincingly. Only a minor suggestion: Concerning their reply that they “did not want to make type-II-errors and missing out on reporting possibly interesting results” I feel like they could even include a statement like this in the limitations sections to make it very transparent to the readers why they decided against corrections.

Respectfully

Sophie Siestrup

7. PLOS authors have the option to publish the peer review history of their article (what does this mean?). If published, this will include your full peer review and any attached files.

Reviewer #1: **Yes: **Sophie Siestrup

---

## [Editor Report · Acceptance letter]

6 Nov 2024

PONE-D-24-24703R1 

PLOS ONE

Dear Dr. Berginström, 

I'm pleased to inform you that your manuscript has been deemed suitable for publication in PLOS ONE. Congratulations! Your manuscript is now being handed over to our production team.

Kind regards, 

on behalf of

Dr. Vincenzo De Luca 

Academic Editor

PLOS ONE